# Immunotherapy in Glioblastoma: A Clinical Perspective

**DOI:** 10.3390/cancers13153721

**Published:** 2021-07-24

**Authors:** Nicolas Desbaillets, Andreas Felix Hottinger

**Affiliations:** 1Department of Clinical Neurosciences, Centre Hospitalier Universitaire Vaudois & Université de Lausanne, 1011 Lausanne, Switzerland; nicolas.desbaillets@unil.ch; 2Department of Oncology, Centre Hospitalier Universitaire Vaudois, 1011 Lausanne, Switzerland

**Keywords:** glioblastoma, immunotherapy, checkpoint inhibitor, vaccine, oncolytic virus, CAR T cell

## Abstract

**Simple Summary:**

Glioblastoma is the most frequent and the most aggressive brain tumor. Even with the most current treatment, its prognosis remains dismal. Immunotherapies, novel cancer therapies using the patient’s own immune system to fight cancer, have revolutionized the treatment of numerous cancer types and generate great hope for glioblastoma. In this review, we analyze the challenges immunotherapy is facing in glioblastoma, present the different immunotherapy approaches with corresponding key clinical trial findings, and finally discuss limitations and how they might be overcome. Proof of efficacy for immunotherapies remains to be demonstrated in glioblastoma, but novel combinatorial approaches remain promising.

**Abstract:**

Glioblastoma is the most frequent and the most aggressive brain tumor. It is notoriously resistant to current treatments, and the prognosis remains dismal. Immunotherapies have revolutionized the treatment of numerous cancer types and generate great hope for glioblastoma, alas without success until now. In this review, the rationale underlying immune targeting of glioblastoma, as well as the challenges faced when targeting these highly immunosuppressive tumors, are discussed. Innovative immune-targeting strategies including cancer vaccines, oncolytic viruses, checkpoint blockade inhibitors, adoptive cell transfer, and CAR T cells that have been investigated in glioblastoma are reviewed. From a clinical perspective, key clinical trial findings and ongoing trials are discussed for each approach. Finally, limitations, either biological or arising from trial designs are analyzed, and strategies to overcome them are presented. Proof of efficacy for immunotherapy approaches remains to be demonstrated in glioblastoma, but our rapidly expanding understanding of its biology, its immune microenvironment, and the emergence of novel promising combinatorial approaches might allow researchers to finally fulfill the medical need for GBM patients.

## 1. Introduction

Glioblastoma multiforme (GBM) is the most frequent malignant primary central nervous system (CNS) tumor in adults [1]. It is highly aggressive, notoriously resistant to all current standard of care treatments, and shows a very poor outcome, with a 6.8% 5-year overall survival [1]. Despite recent advances in our understanding of glioblastoma (GBM), therapeutic progress remains desperately needed. It is believed that tumor heterogeneity and the tumor microenvironment limit GBM’s sensitivity to standard of care approaches. Immunotherapy has now demonstrated its efficacy against a wide range of solid tumors, including melanoma, non-small-cell lung cancer, and renal cell carcinoma, establishing the 5th pillar of anticancer treatment [2]. Unfortunately, this progress has not yet translated to improved outcome for glioblastoma patients. Despite this, there are many incentives to use immunotherapy to treat glioblastoma, and in this review, focusing on randomized clinical trials, we will discuss the rationale of such approaches, their current status and limitations. We will also take a look at future outlooks and provide elements to optimize clinical trials in immunotherapy.

## 2. Standard of Care

In newly diagnosed GBM, the standard of care consists of maximum safe tumor resection, followed by radiotherapy (RT) and concomitant temozolomide (TMZ) chemotherapy. This combined treatment offers a 14.6- vs. 12.1-month median overall survival (OS) compared to the radiotherapy alone. MGMT promoter methylation is a predictive factor for better outcome [3]. In 2015, the FDA granted approval to a novel electro-physical treatment modality, tumor-treating fields (TTFields) for GBM patients. The phase III (NCT00916409) trial has demonstrated improved median progression-free survival (PFS), with 6.7 months in the TTFields-temozolomide group versus 4.0 months in the temozolomide-alone group. Median OS was also significantly improved, with 20.9 months vs. 16.0 months (HR, 0.63; 95% CI, 0.53–0.76; *p* < 0.001) [4].

However, virtually all GBMs will relapse. The management of progressing or relapsing tumors is typically more individualized than the standard first-line therapy, and accounts for patient-specific factors such as the time since diagnosis, previous treatments, and most importantly, the patient’s performance and neurological functions. The available therapeutic options include second-line surgery, radiotherapy, chemotherapy with alkylating agents, and antiangiogenic therapy with bevacizumab [5]. Unfortunately, from the first progression or recurrence onward, the median OS ranges only from 6 to 9 months [6]. As a consequence, novel treatment strategies are urgently needed to treat recurring GBM. Uncountable clinical trials have addressed this highly unmet medical need for GBM treatment. Over 100 different targeted drugs have been investigated to date, however without clinical benefits [7], thus leveraging the hopes on immunotherapy. It is noteworthy that recurrent GBM previously treated with radiotherapy and chemotherapy typically present higher mutational burdens and are expected to be more immunogenic than their untreated counterparts, further fostering faith and optimism in immunotherapy for relapsing GBM patients. The interaction between the immune system and the cerebral parenchyma presents a number of specificities that may complicate this goal and must be addressed.

## 3. Immune Privilege of the CNS

The central nervous system (CNS) has long been considered an immune privileged system: Due to the blood–brain barrier (BBB) blocking access to pathogens, the CNS is far less exposed to blood-borne pathogens than any other organ [8]. Evolutionarily, with the infrequent need to mount immune attacks and the elevated consequences of auto-immune aggressions against brain cells, dampening immunity in the CNS was probably advantageous. Until 2015, it was believed that the CNS lacked functioning lymphatics. As those are an essential component of the immune response, it was therefore difficult to understand how antigen presentation could occur [9]. The BBB itself is also considered a limiting factor for an efficient immune response, as its tight junctions physically block the entry of immune players such as lymphocytes or antibodies [10]. Furthermore, a key difference between the CNS and other organs lies in the quasi-absence of dendritic cells for antigen presentation in the brain [11]. In the CNS, microglia is considered the main antigen presenting population, and it adopts an anti-inflammatory phenotype and skews T cells to an immunosuppressive Th2 phenotype [12,13]. However, it is now well demonstrated that active immunosurveillance does occur in the CNS, and that efficient immune responses are mounted in response to infections (reviewed in [14]). Moreover, autoimmune diseases such as multiple sclerosis also show that immunogenic antigens can be processed and trigger robust immune responses in the CNS. In 2015, the identification of lymphatic pathways along dural venous sinuses leading to the deep cervical lymph nodes greatly changed our conception of the brain’s immune environment [9]. Today, while the CNS is considered an immunologically distinct site, it is believed that its immune microenvironment offers appropriate conditions for immunotherapy targeted toward brain tumors (Figure 1).

## 4. Mechanisms of Immune Evasion by GBM

The fact that GBM is the deadliest form of brain cancer, with rapid growth and frequent relapse, can be attributed to several factors, including a high proliferation rate, notorious tissue-invasion capacity, treatment-resistant cancer stem cells, and difficult access of drugs to the CNS. In addition to these, immune evasion plays a key role in the poor outcome of GBM. A number of mechanisms of immune evasion have been identified, including prevention of entry of immune cells by an intact blood–brain barrier, immune suppression by the tumor microenvironment, or modulation of the immune system by hijacking key immune pathways and players such as immune checkpoint receptor expression, regulatory T cells, and tumor-associated macrophage modulation. These mechanisms of immune evasion used by GBM are the subject of intensive research, and an in-depth review goes beyond the scope of this article, but have been thoroughly reviewed by Razavi et al. [15]. Targeting GBM with immunotherapy implies understanding its immunosuppressive mechanisms and reversing them.

GBM possesses high intrinsic resistance mechanisms as well as an impressive capability to adapt to immune attack, and these responses are only transient due to acquired resistance mechanisms. One study investigating PD-1 blockade in GBM showed only few patients with an initial response, and all of them relapsed. Pathology on relapsed tumor biopsies demonstrated novel expression of immunosuppressive molecules and loss of neoantigen expression [16].

Intrinsic resistance mechanisms are characteristics that arise from the location, tissue of origin, and basic tumor biology of GBM. First of all, GBM gains immunosuppressive properties from its location in the CNS [17,18]. Preclinical studies have shown that the intracranial location is by itself sufficient to induce systemic immunosuppression against its antigens. For instance, brain tumors from B16 melanoma cells expressing the Pmel model tumor antigen induced systemic immune tolerance by deleting and blocking cytotoxic responses. This phenomenon was specific to the brain, as it was not observed when identical tumors were implanted in the lungs or flank [17].

GBMs also gain advantages from their vast intratumoral cellular heterogeneity. One study analyzed biopsies arising from different tumor regions across 11 patients and found significant molecular changes across different areas of the same tumor in every patient [19]. This heterogeneity plays a fundamental role in tumor adaptability and resistance to treatment with rapid outgrowth of clonal populations that become treatment-resistant [20]. Alkylating agents such as temozolomide, given their intrinsic mechanism of action, have been reported to induce recurrences with increased tumor mutational load in about 10% of patients [21]. These hypermutant recurrent gliomas may increase this tumor adaptability, but may also present unique molecular vulnerabilities with the development of multiple additional tumor-specific neoantigens.

This selective pressure applies to classical growth-signaling pathways and DNA repair mechanisms, but also to neoantigens. Immune editing is a three-part concept described by Dunn et al. [22] and includes a continuum of elimination, equilibrium, and escape. This concept reasons that immune surveillance constantly eliminates precancerous clones and exerts evolutionary pressure, selecting the least-immunogenic clones and ultimately leading to equilibrium and further escape from homeostatic immune surveillance. It is postulated that this mechanism allows GBM to evolve and counter-adapt to immune attacks, mostly by usurping mechanisms that normally guard against autoimmunity. Preclinical studies have demonstrated that GBM intra-tumoral T cells are scarce and express multiple immune checkpoints, leading to a severe exhaustion signature [23]. Patients with hypermutated GMB tumors treated with anti-PD1 checkpoint inhibitors appear to transiently respond to treatment, but ultimately escape [24,25]. Upregulation of additional checkpoints is believed to contribute to this resistance and concurrent administration of PD-1 and TIM-3-blocking antibodies synergies in preclinical models [26]. Due to this high adaptability, immunotherapy strategies should aim for neoantigens matching certain quality criteria such as being expressed across numerous subclonal populations, being absent in normal tissue, and ideally on which tumors cells depend on for growth and survival to avoid immune editing.

GBM also benefits from a favorable microenvironment and even tailors it further to be immunosuppressive [27]: As mentioned, microglia constitute the vast majority of the myeloid compartment in the CNS, and they function as the main APC population. Although microglia express major histocompatibility complex (MHC) class II molecules and present antigens to activated lymphocytes, they do not efficiently prime naive lymphocytes [12,28]. Instead, microglia tolerize naive lymphocytes once they enter the CNS [29]. One study showed that CD8+ T cells primed against CNS antigens were rapidly deleted upon entry in the CNS, demonstrating the tolerizing role of the CNS microenvironment [30]. The BBB guards the entry of immune cells. However, in an inflamed context, interferon-inducible chemokines activate endothelial cells and allow peripheral immune cells to cross the BBB [31]. GBM skews this mechanism by upregulating chemoattractant proteins in the stroma, and recruiting monocytes from the periphery. Among these monocytes are myeloid-derived suppressor cells [32] and tumor associated macrophages (TAMs) [33]. TAMs are key players in the tumor stroma, as they have been shown to sustain genetic instability, promote epithelial-to-mesenchymal transition (EMT), support cancer stem cells, and promote the expression of immune checkpoint ligands [34]. GBM, via cellular metabolites, recruits TAMs and drives their polarization into anti-inflammatory ‘M2′ macrophages [35]. TAMs also regulate the T-cell composition in the tumor microenvironment. One retrospective study analyzing 284 gliomas of different grades identified a high CD4+/CD8+ ratio as predictive of poor overall survival, and regulatory T cells (Tregs) being present in high-grade but not in low-grade gliomas [36]. Directly targeting TAMs is being investigated either by blocking their recruitment, inhibiting their immunosuppressive function, or even reprograming them to a tumoricidal ‘M1′ phenotype. CSF-1 (colony-stimulating factor-1) being critical for TAMs function and survival, trials mostly rely on its inhibition, but so far without clinical benefit [37,38]. Along the same lines, one ongoing clinical trial investigated IDO inhibitors in glioblastoma (NCT02052648). In addition to reprogramming TAMs, IDO inhibitors also had effects on Treg cell accumulation [37,39]. Tregs can also be targeted with monoclonal antibodies against the glucocorticoid induced TNFR-related protein (GITR), which has shown interesting benefits in a preclinical mouse GBM model [40].

Finally, one barrier to immunotherapy in GBM is iatrogenic immunosuppression. Radiation therapy with concomitant temozolomide chemotherapy is the standard of care in GBM [41]. One study showed that this treatment induced a drop in CD4+ T-cell counts below 300 cells/mm^3^ in 3/4 of patients [42]. In addition, temozolomide prevented the induction of memory T cells in PD-1 blockade preclinical trials [43]. These immunosuppressive effects, combined with the frequently prescribed anti-inflammatory corticosteroids [44] to reduce edema, increase the complexity when developing immunotherapies and raises questions about their integration with the standard of care in the trial designs.

Treating GBM constitutes a challenge of overcoming various resistance mechanisms. Appropriate target epitopes must be identified, and the immunosuppressive microenvironment secondary to CNS tumor location must be overcome while iatrogenic immunosuppression must be minimized. Immunotherapy carries great hopes for GBM because of its potential to overcome these challenges. Activated T cells screen for their targets through healthy tissues with great specificity and can extravasate into brain tumors [45]. Epitope spreading (a process in which the antitumor immune response broadens and novel epitopes distinct from and non-cross-reactive with the initially targeted tumor epitope become additional targets) allows the T-cell repertoire to adapt to the tumor’s evolving molecular profile [26]. Finally, B- and T-cell immune memory is believed to prevent relapse [43].

## 5. Immunotherapy Strategies

### 5.1. Vaccine Approaches

Anticancer immunotherapy began over a century ago with William Coley’s toxin leading to tumor shrinkage when injected intratumorally. Since then, our understanding of immunology has greatly expanded. We now understand the mechanisms at stake and learned to design anticancer vaccines aimed at inducing cytotoxic cellular immune responses capable of eradicating tumor cells. The development of a GBM vaccine also nicely illustrates the achievements reached so far, highlights the current limits to the approach, and provides key elements on the remaining steps that need to be overcome.

Like preventive vaccines used against infectious diseases, anticancer vaccines consist of tumor antigens injected with an adjuvant in the hope of triggering and boosting an immune response. Many variants of this approach have been developed over the years and are still being investigated. In terms of antigen selection, it remains unclear whether whole tumor lysates offering a wide range of neoantigens are superior to in vitro selected potent and specific tumor antigens. The delivery method is also subject to intensive research. Three main approaches have been considered in GBM: (1) peptide/DNA vaccines involve the injection of tumor-specific antigens or nucleic acids, often with immune stimulatory molecules to improve the adaptative immune response; (2) in cell-based therapies, peripheral blood mononuclear cells (PBMCs) can be differentiated into mature dendritic cells that are then primed and loaded with tumor antigens prior to being reinfused to the patient; and (3) alternatively, viral vectors loaded with mRNA coding for key tumor antigens can be used as vaccination platform triggering potent immune responses (reviewed in [46]; Figure 2).

One of the main concerns of vaccine development has been to minimize the risk of off-target toxicities in the CNS. For this reason, one of the first and most evaluated vaccine approaches has concerned the alternatively spliced variant III (vIII) of EGFR, which is a tumor-specific antigen resulting from alternative splicing of exons 2 to 7. EGFRvIII is expressed in 25–30% of GBM tumors. Several vaccine approaches have been developed targeting EGFRvIII (Table 1).

Rindopepimut (CDX-110) has been the most extensively studied EGFRvIII peptide vaccine. It uses the immunomodulatory keyhole limpet hemocyanin (KLH) as an adjuvant, and was recognized as “breakthrough therapy” by the FDA for GBM in February 2015, based on promising preliminary, nonrandomized phase II data that showed improved PFS and OS compared to 19 historical matched controls, as well as an elimination of EGFRvIII in some vaccinated patients.

A randomized phase III ACT IV trial further investigated the product [47]. After tumor resection and radiochemotherapy, patients were randomized to receive either temodal and rindopepimut or temodal and placebo. This trial was discontinued early because of lack of improvement in OS, the primary endpoint of the study (HR = 1.01; 95% CI: 0.79–1.30; *p* = 0.93) [47]. Moreover, the number of patients that showed a loss of EGFRvIII expression was identical in both groups, confirming that this is an event that occurs spontaneously in about 25% of GBM patients [47]. Importantly, these results were observed despite the demonstration of the development of robust antigen-specific antibody response. These humoral immune responses are usually taken as surrogate markers for immunogenicity and are considered a prerequisite for a therapeutic impact. This humoral response, however, does not evaluate whether a specific T-cell response could be generated, and does not address which MHC class I or II epitopes are required for CD8+ or CD4+ T-cell priming. This information would be essential to determine the treatment chances of ongoing development of T-cell-based immunotherapies targeting EGFRvIII with CAR T cells or bispecific antibodies.

The selection of a single target antigen might be a significant contributor to the lack of efficacy of this approach. Even though EGFRvIII is exclusively expressed on GBM tumor cells, only a fraction of the cells expresses this abnormal antigen [48,49]. Therefore, a significant therapeutic effect can only be achieved if these cells represent significant tumor stem cells that will drive the extinction of the cancer if they are eliminated, or if their elimination results in a significant modification of the tumor microenvironment to subsequently allow the immune system to target other cancer cells.

To increase the odds of developing an efficient immune response against GBM, a number of steps have been taken. A first approach was to identify multiple tumor-specific antigens and combine them within one vaccine. IMA950, a peptide vaccine that combines 11 GBM-derived antigens that are commonly found in GBM, represents one of these approaches. This vaccine has been shown to elicit T-cell responses to single and multiple antigens [50]. However, randomized trials have not yet been performed, so whether this immune response indeed leads to improvement in the outcome remains to be demonstrated. A second step was to identify so-called neoantigens. These antigens represent structural genomic abnormalities resulting from DNA alterations within the tumor. These neoantigens can be identified by whole-exome DNA and RNA sequencing and are selected based on their predicted affinity to bind to specific MHC molecules using computational algorithms [51]. Accumulating data suggest that neoantigens represent an important therapeutic target for cancer therapy: higher neoantigen loads have been correlated with improved outcome in melanoma patients treated with immune checkpoint inhibitors [52].

Dendritic cells (DCs) are powerful antigen-presenting cells able to induce antigen-specific T-cell responses. DCs can be isolated from the patient and stimulated ex vivo against tumor-specific antigens and then reinjected into the patient to reinforce the immune response against the tumor. To date, two randomized trials have evaluated the use of dendritic vaccines: ICT-107 is an autologous DC immunotherapy targeting six antigens of tumor and cancer stem cells: melanoma-associated antigen-1 (MAGE-1), antigen isolated from melanoma-2 (AIM-2), human EGFR-2 (HER2/neu), tyrosine-related protein-2 (TRP-2), glycoprotein 100 (gp 100), and IL13 receptor alpha 2 (ILRa2) identified in the tumor tissues of 46 GBM patients. In a double-blind, placebo-controlled trial involving 124 patients, 75 patients were treated with ICT-107 following RT and concomitant TMZ, and 42 were included in the placebo arm (DC-pulsed without tumor-specific antigens). Whereas median PFS was slightly superior for the treatment group (11.2 vs. 9 months; HR:.57, *p* = 0.011), there was, however, no difference in OS (17.0 vs. 15 months; HR: 0.87, *p* = 0.580). Interestingly, patients with a demonstrated immune response (using IFNg ELISpot) showed an improved PFS and OS compared to non-responders. Moreover, in this trial, control patients also showed a high immune response rate in the control population (who received activated and mature DCs), and may have mounted an immune response by processing free tumor antigens at the lymph nodes to prime T cells, including some of the antigens selected for this trial. Preliminary results of a large-scale, randomized, and controlled phase III trial of the dendritic cell-based vaccine DCVax-L in newly diagnosed GBM patients were published recently (NCT00045968) [53]. In this trial, patients were randomized 2:1 to standard of care plus DCVax-L (*n* = 232) or standard therapy plus placebo (*n* = 99). Patients received intradermal injections of DCVax-L every 2 months for the first year and then every 6 months. Contrary to usual practice, the authors reported the results of both combined treatment and control arms of their trial as a single arm, arguing that nearly 90% of the ITT population received DCVax-L either as they were enrolled in the treatment arm, or because they were offered compassionate access after progression. Because of this, PFS data have not yet been reported. The authors focused on the median OS of 23.1 months post-surgery as the indication of efficacy compared to historical control groups of similarly designed clinical trials [4,47]. It is important, however, to note that the OS of those trials was presented from time of randomization rather than diagnosis (as done in the DCVax trial), and once corrected for this factor, the OS confidence intervals of the control groups of the two trials and the DCVax population overlapped. Moreover, importantly, in this trial, out of 1599 screened patients, only 331 were actually randomized, raising the possibility of a selection bias toward patients with a more favorable outcome [54]. It therefore remains to be verified that this trial, which started in 2007, will confirm a superior outcome of the treatment arm.

More recently, a randomized phase II trial in 72 relapsed GBM patients investigated the combination of rindopepimut and bevacizumab (ReACT trial) in patients with recurrent GBM. The primary endpoint, PFS at 6 months, was 28% (10/36) for the treatment arm, compared to 16% (6/37) for the control intervention (*p* = 0.12). Moreover, median OS was significantly improved in patients receiving the combined treatment (HR = 0.57 (0.33, 0.98), *p* = 0.039) [55]. These findings will need validation in a larger randomized trial.

A number of candidate vaccines are still under development: Among them, the NCT02287428 trial, a cohort of eight patients with high-risk MGMT unmethylated GBM, received standard of care treatments and the peptide-based vaccine containing up to 20 long peptides, each containing 3–5 personalized neoantigenic peptides. These neoantigens were chosen by whole-exome sequencing (WES) of patients’ resected tumor material and HLA class I binding predictions. Interestingly, the authors demonstrated that the vaccine failed to elicit robust T-cell responses in patients that received dexamethasone, a steroid immunosuppressant commonly used to decrease swelling and edema around cerebral lesions. Two patients who had not received steroids exhibited a specific T-cell response with CD4+ and CD8+ T-cell infiltration. Neoantigen-specific T cells expressed degranulation and memory surface markers. However, despite these promising results, all the vaccinated patients, even the two who responded, ultimately died of the disease. One key finding of this study relied on the analysis of relapsing tumor material in one of the initially responding patients. Neoantigen-specific T cells found in the tumor expressed exhaustion markers, rendering them incapable of recognizing and killing target cells. This exhausted state can be reversed by immune-checkpoint inhibitors [56]. This trial is still actively recruiting, and additional cohorts are currently being investigated in combination with pembrolizumab. Also, a subcohort of MGMT-methylated patients is being added. Along the same lines, the pilot study NCT03422094 trial is investigating NeoVax in combination with ipilimumab or nivolumab.

A phase I trial tested an interesting combination of peptide-based vaccines: GAPVAC-101 (NCT02149225) included 15 GBM patients who were treated, in addition to the standard of care, with a vaccine (APVAC1) based on a premanufactured library of 30 unmutated GBM overexpressed antigens followed by vaccination with APVAC2, a personalized vaccine that targeted patient-specific neoepitopes. Personalization was based on WES, transcriptomics, and immunopeptidomes of the patients’ individual tumors. Interestingly, APVAC1 antigens elicited central memory CD8+ T cells, whereas APVAC2 induced predominantly Th1 CD4+ T-cell responses [57]. The median OS was 29 months [57]. This trial tended to prove that vaccination is an attractive approach in patients with GBM, but due to small patient numbers needs further validation in larger randomized cohorts.

### 5.2. Oncolytic Viruses

Oncolytic viruses (OVs) comprise a diverse group of biologic agents with potential as cancer therapeutics. OVs are viruses engineered to selectively infect cancer cells and act intratumorally. Initially, they were developed to directly lyse cancer cells. Today, with our understanding of immunotherapy, OV’s potential applications have broadened. Their lytic activity is seen as an opportunity to release tumor antigens and disrupts the extracellular matrix and tumor microenvironment architecture that can favor T-cell infiltration and the release of tumor-specific antigens [58]. Moreover, oncolytic viruses may activate the adaptive immune system through activation of toll-like receptor and pathogen-associated molecular pattern sensors and stimulate dendritic cells to produce type I interferons, resulting in a proinflammatory immune response, and production of CXCL9, CXCL10, and CXCL11 cytokines that, in turn, activate T-cell trafficking and local infiltration [59]. Furthermore, OVs are used as vectors to deliver signals and genetic material intratumorally. Payloads typically include genetic material to synthetize potent neoantigens for subsequent immunotherapy or key cytokines to regulate the immune environment. The exceptional results in a phase III advanced melanoma trial (NCT00769704) [60] using this approach, as well as the subsequent FDA and EMEA approval of the engineered immunostimulatory OV talimogene laherparepvec, brought OVs in focus recently [61,62,63].

Several OV approaches, including adenovirus, measles, polio, HSV, parvovirus, and replicating retroviral vectors, have been investigated in GBM and proved the feasibility and safety of the approach. Recently, novel OVs in phase I/II GBM trials suggested remarkable efficacy with subsets of patients achieving over 3 years of survival. These include adenovirus DNX-2401 (Ad5-delta24-RGD) [64], measles virus MV-CEA [65], parvovirus H-1 (ParvOryx) [66], polio-rhinovirus chimera (PVSRIPO) [67], and the retroviral vector Toca 511 (vocimagene amiretrorepvec and Toca FC) [68]. To date, however, no randomized trials are available to confirm these findings suggestive of efficacy [69] except for Toca 511, which underwent evaluation in a phase III trial: the murine leukemia virus-based Toca 511 (vocimagene amiretrorepvec) is a retroviral vector encoding the yeast cytosine deaminase that converts 5-fluorocytosine (5-FC), an antifungal drug, into the antimetabolite drug 5-fluorouracil. A phase I trial (NCT01470794) tested injecting Toca 511 into the resection cavity of 56 recurrent high-grade gliomas patients, followed by cycles of 5-FC [68]. In the patient subgroup matching the follow-up phase III study eligibility criteria (23 patients), which included both wild-type and IDH1-mutant tumors, median OS was 14.4 months, and OS at 1 and 2 years was 65.2% and 34.8%, respectively. Five patients demonstrated complete response and have been alive 33.9–52.2 months after Toca 511 administration [68]. The Toca 5 trial, a multicenter, randomized, open-label phase II/III trial of Toca 511 combined with 5-FC versus standard of care in 403 patients undergoing planned resection for recurrent glioblastoma or anaplastic astrocytoma was terminated in 2020 due to lack of efficacy (NCT02414165). The median OS was 11.10 months for the Toca 511/5-FC group and 12.22 months for the control group [70].

DNX-2401 (Ad5-Delta-24-RGD; tasadenoturev) is a replication-competent, infectivity-enhanced, tumor-selective oncolytic adenovirus 5 (Ad5)-based vector [71]. Tumor-cell targeting is achieved via the deletion of 24 base pairs in the E1A protein and the insertion of an Arg–Gly–Asp (RGD) motif in a viral capsid protein, increasing the affinity for αV integrins [72]. A total of 37 patients with recurrent malignant glioma were included in the DNX-2401 dose-escalation trial and received intratumoral injection of the OV; 20% of patients survived over 3 years post-treatment, and three of those patients had a PFS over 3 years. In a second cohort, post-treatment biopsies revealed that DNX-2401 can replicate within the tumor and induce potent intratumoral CD8+ and T-bet+ T-cell infiltration, but this occurs only in a subset of patients. DNX-2401 is safe, as no dose-limiting toxicities have been observed [64]. The phase Ib (TARGET-I trial, NCT02197169) was a randomized trial testing the OV alone or combined with interferon gamma (IFN-γ) in recurrent GBM. Although the addition of IFN-γ did not improve survival, the combined 1- and 1.5-year OS in both groups was 33% and 22%, respectively [73]. Another phase II combination trial investigating intratumoral injection of DNX-2401 with systemic administration of anti-PD-1 checkpoint inhibitor pembrolizumab in 48 recurrent GBM patients is ongoing (CAPTIVE/KEYNOTE-192, NCT02798406). Interim results tend to show that long-term survival and clinical benefit remain compelling, as the median OS was 12 months, OS at 6 months was 91%, but only 47% of patients experienced a clinical benefit (stable disease or regression). Four patients had a partial response, and only three were alive >20 months [74].

Other adenoviral vectors are being investigated. Phase I trials (NCT02026271 and NCT03330197) are testing intratumoral injection of Ad-RTS-hIL-12, an inducible adenoviral vector expressing human interleukin 12 (hIL-12) in the presence of the activator ligand veledimex. The NCT02026271 trial is a dose-escalation trial performed in 38 adult patients with recurrent or progressing high grade gliomas [75] that showed a favorable safety profile and mitigated survival (median OS of 12.7 months). The NCT03330197 trial is its pending pediatric trial that is still recruiting, and is aiming for 25 pediatric patients with recurrent or progressing high-grade glioma.

The Edmonston vaccine strain of measles virus is a safe and specific oncolytic virus that has been genetically modified to express human carcinoembryonic antigen (CEA) as a reporter gene to monitor viral replication in vivo [65]. The MV-CEA OV has been tested in a phase I clinical trial (NCT00390299) in 23 GBM patients. MV-CEA was either injected intratumorally prior to en block tumor resection or injected in the resection cavity. Median OS was 11.4 and 11.8 months, respectively, and the median PFS at 6 months was 22–23% in both groups.

A modified rat parvovirus named oncolytic H-1 parvovirus (ParvOryx) was tested in a phase I/IIa dose-escalating trial in 18 patients with recurrent GBM (NCT01301430) [66]. An initial dose of ParvOryx was administered either intratumorally or iv and 9 days later, a second dose was administered around the resection cavity. Median OS was 15.5 months after first ParvOryx treatment. Eight patients survived >12 months and three patients >24 months after the first administration of ParvOryx. Analysis of tumors biopsies revealed that strong CD8+ and CD4+ T lymphocytes infiltration occurred in six patients [66]. Clinical response was independent of the administration route. In terms of safety, no maximum tolerated dose could be identified.

In May 2016, the recombinant oncolytic poliovirus PVSRIPO received breakthrough therapy designation from the FDA on the basis of the findings of an ongoing phase I study in patients with recurrent glioblastoma. The polio–rhinovirus chimera (PVSRIPO) targeting the poliovirus receptor CD155 is an engineered replication-competent attenuated Sabin type 1 poliovirus with its internal ribosome entry site replaced by a human rhinovirus type 2 sequence to eliminate its neurovirulence. A phase I dose-escalation trial (NCT01491893) was conducted in 61 recurrent supratentorial grade IV malignant glioma patients receiving intratumoral infusion of PVSRIPO [67]. Safety is debated, as 19% of the patients had grade 3 or higher adverse events, although no neuropathogenicity or virus shedding was observed. In terms of survival, median OS among all 61 patients was 12.5 months. This encouraging result must be pondered, however, due to the fact that the inclusion criteria were limited to patients with limited tumor size and absence of steroid requirement. Nevertheless, the outcome compared favorably to a cohort of historical controls that matched the inclusion criteria of the trial. Moreover, interestingly, about 20% of patients remained alive for 57–70 months after the PVSRIPO infusion [67]. Based on these results, PVSRIPO received breakthrough therapy designation from the FDA in May 2016, and a randomized phase II trial of PVSRIPO alone or combined to a single cycle of lomustine in patients with recurrent grade IV malignant glioma (NCT02986178) is ongoing.

Several other clinical trials in adult patients with recurrent glioblastoma/glioma are ongoing, such as a phase I/II trial of the vaccinia-based OV TG6002 combined with 5-FC (ONCOVIRAC, NCT03294486); a phase I trial of the adenoviral bases OV DNX-2440 expressing the immunostimulatory OX40 ligand (OX40-L) (NCT03714334); a phase I trial of a genetically engineered herpes simplex virus (HSV-1) expressing IL-12 named M032 (NCT02062827); and a phase I trial of a genetically engineered HSV-1, rQNestin34.5v.2, combined with cyclophosphamide (NCT03152318).

In general, these early-phase clinical trials suggest that OVs can improve survival in subsets of patients. However, a metanalysis of oncoviral trials addressing recurrent GBM revealed that the 2- and 3-year survival rates were not statistically different to standard of care (15% versus 12% at 2 years; 9% versus 6% at 3 years) [76]. Although promising, the benefit of oncolytic virotherapy remains yet to be proven in large randomized controlled phase II/III trials.

### 5.3. Checkpoint Inhibitors

Checkpoint inhibitors (CPIs) are monoclonal antibodies targeting surface receptors called immune checkpoints regulating key molecular signaling pathways dampening the activation of T-cells. Among the common targets are Programmed Death 1 (PD-1) and its ligand (PD-L1), to overcome the inhibition of T lymphocytes by tumor cells. These anti-PD-1 and anti-PD-L1 antibodies have revolutionized the treatment of many tumor types, including melanoma, lung cancer, and kidney cancer [77,78,79]. The expression of PD-L1 in certain tumor types is a predictive factor of response to anti-PD-1 or anti-PD-L1 therapy; 88% of newly diagnosed glioblastoma and 72% of relapsed glioblastoma show PD-L1 overexpression, although generally at a low level [80]. Levels of PD-L1 expression also seem to correlate with the severity of the prognosis [81].

In general, the tumor mutation load is associated with a better response to immune CPIs [82]. GBM, however, typically exhibit a low mutation burden compared to other tumor types such as melanoma or lung cancer [83]. Colon cancer typically shows poor responses to CPIs, but a subset of tumors with microsatellite instability and an accordingly higher mutation burden responds well to CPIs. The combined analysis of tumor mutational loads, mismatch repair (MMR), and immune checkpoint expression in 198 GBM patients revealed that only 3.5% of them showed high mutational burden associated with the loss of MLH1, MSH2, MSH6, and PMS2 expression [84]. Interestingly, other studies have demonstrated the acquisition of MMR deficiency in relapsed GBM. Acquired mutations in MSH6 after temozolomide and radiotherapy treatment occurs in up to 25% of patients [85,86,87]. Thus, there is a rationale of testing CPIs in relapsed glioblastoma, but it is expected that only few GBM patients (mainly those with DNA repair deficiency) may show clinical benefit.

Anti-PD-1 CPIs in relapsed GBM has been the subject of many phase I clinical trials (NCT02017717, NCT02336165, NCT02337491, NCT02054806) (Table 1) shows response rates ranging from 2.5 to 13.3% with anti-PD-1 or anti-PD-L1 monotherapy. Progression-free survival (PFS) rates at 6 months ranged from 16% to 44%, and the overall survival (OS) between 7 and 14 months. In phase I, the CheckMate-143 (NCT02017717) trial investigated the efficacy of nivolumab (anti-PD-1) alone or in combination with another CPI, ipilimumab (anti-CTLA4), in patients with a first GBM relapse. Various combinations of CPIs and dosing regimens were tested but, in fine, nivolumab alone (3 mg/kg body weight every 2 weeks) showed a greater median OS (10.4 months vs. 9.2 and 7.3 months) than the combination regimens and less toxicity. Double-checkpoint inhibition elicited severe adverse events in over 50% of patients, and this approach is no longer pursued [88]. Phase III of the CheckMate-143 (NCT02017717) trial comparing nivolumab 3 mg/kg (*n* = 184) to bevacizumab (anti-VEGFA MAb) 10 mg/kg (*n* = 185) has been terminated early due to lack of efficacy. No statistical difference was noted in terms of median OS, nor toxicity in both groups. However, responders to nivolumab (7.8%) had a sustained response over time: the median duration of radiologic response was 11 months in the nivolumab arm versus 5.3 months with bevacizumab in responders. [89,90]. Two ongoing phase III trials are investigating nivolumab for the treatment of newly diagnosed GBM: The CheckMate-548 trial is testing temozolomide plus radiotherapy combined with nivolumab or placebo in patients with newly diagnosed MGMT-methylated GBM (NCT02667587), and the CheckMate-548 trial is investigating nivolumab versus temozolomide, in combination with radiotherapy, in patients with newly diagnosed MGMT unmethylated GBM (NCT02617589). Information available so far suggests that neither trial will reach its primary endpoint.

The Keynote-028 trial (NCT02054806) investigated the efficacy of pembrolizumab, another anti-PD1 CPI, in several advanced solid tumors, including 26 glioblastoma patients. Among them, one patient had a partial response (4%), and 12 patients (48%) showed stable disease according to iRECIST criteria, with a median PFS of 2.8 months and a median OS of 14.4 months [91]. A phase II trial investigated the benefit of the anti-PD-L1 CPI durvalumab, as monotherapy or combined with bevacizumab or radiotherapy in 30 relapsed glioblastoma patients. Four patients (13.3%) had a partial response, and 14 patients (46.7%) presented stable disease. The PFS at 6 months was 20% [92].

In GBM preclinical trials, CPIs appeared promising, but these findings could not be translated in larger randomized trials so far, highlighting the need to better identify predictive factors of response in order to stratify eligible patients for CPI therapy.

### 5.4. Adoptive Cell Transfer and CAR T Cells

Adoptive T-cell transfer (ACT) consists of re-infusing a patient their own (autologous) or donor (allogenic) antitumor T cells to attack tumor-specific antigens on the patient’s cancer cells. This approach reproduces classical expansion and activation of antigen-specific T-cells. However, in ACT this expansion is performed ex vivo. Tumor infiltrating lymphocytes (TILs) are isolated from a tumor resection piece, then activated and expanded in vitro prior to being reinfused into the patient with systemic IL-2 stimulation after preparative lymphodepletion. This maximizes the specific T-cell numbers a tumor faces and ensures a proper activation state of the T cells that are thus less susceptible to the intratumoral immunosuppressive microenvironment. This treatment approach demonstrated spectacular clinical results with complete responses in advanced metastatic melanoma patients that were previously refractory to other treatments [93]. Furthermore, adoptive transfer of TILs in metastatic melanoma patients was effective in 30% of patients refractory to anti-PD1 treatment and 25% of patients refractory to CTLA4 inhibition [94].

In GBM, one pilot trial investigated autologous TIL infusion in six glioma patients [95]. Three patients had anaplastic astrocytomas, and three patients GBM. Among the 3 GBM patients, two showed a partial response. In another pilot trial, GBM patients treated with lymphokine-activated killer (LAK) cells, had a better prognosis compared with contemporary GBM patients [96]. LAK cells are generated by culturing of peripheral blood mononuclear cells (PBMCs) in the presence of IL2, and then cultured with CD3 and OKT3 antibodies to enhance cell expansion [97]. LAK cells are thus a heterogeneous cell population mainly composed of natural killer cells (NK) and natural killer T cells (NKT), which display unrestricted MHC antitumor activities [98]. One clinical trial investigating the effect of LAK cells in GBM was performed (NCT00331526). A total of 33 patients received adjuvant LAK cells infusion in the tumor resection cavity. The median OS of these patients from diagnosis of GBM was 20.5 months [99]. These results supported the feasibility of TIL in GBM, but, given the fact that most patients progressed after treatment, the clinical benefit remains limited. The success of TILs therapy in GBM may be hindered by the difficulty to identify tumor-reactive T cells. For TILs to function, it is necessary to sort and amplify the most tumor-reactive clones. Unfortunately, validated tumor-reactivity markers are still lacking, especially in GBM. Another difficulty arises from the exhaustion state of isolated T cells. Although TILs isolated from GBM tumors have the advantages of tumor specificity and broad antigen repertoire [100,101], they are found in a severe exhaustion state [23,101]. These exhaustion molecules remain highly expressed in amplified TILs [102]. Checkpoint inhibitors may leverage the antitumor efficacy of TIL therapy as until now, the lack of potent tumor-reactive TILs in GBM hinders the clinical development of this strategy [103]. A very recent preclinical study demonstrated that terminally exhausted TILs could be reverted by metabolic reprograming using modified and stabilized IL-10, and reversed TILs could eradicate melanoma and colon cancer tumors [104]. We look forward to seeing this approach being tested in GBM clinical trials, hopefully unleashing TILs against GBM tumors.

In the absence of potent T-cell clones, one elegant approach is TCR engineering, in which TCRs are either humanized from high-affinity murine TCRs or screened by yeast display [105]. To date, no clinical trial with TCR-modified T cells has been initiated for GBM. Recently, an HLA-A2 specific TCR recognizing the histone 3 variant 3K27M (H3.3K27M) mutation was identified. H3.3K27M is frequently expressed in diffuse intrinsic pontine gliomas (DIPG) and is shared in 70% of diffuse midline gliomas. In a preclinical model of DIPG, engineered T cells with this TCR inhibited the progression of human xenografts, providing a strong basis for developing this approach in clinical trials [106].

An emerging branch of the immunotherapy arsenal uses gene therapy to engineer and optimize T lymphocytes with a novel chimeric antigen receptor. These chimeric antigen receptor T cells (CAR T cells) are a game-changer in the treatment of various hematological malignancies such as acute lymphoblastic leukemia (ALL) or diffuse large B-cell lymphoma (DLBCL) [107,108]. CAR T cells consist of patient autologous T cells, engineered in vitro with a lentiviral vector allowing them to stably express a chimeric receptor capable of recognizing tumor cells in an MHC-independent manner while simultaneously activating the T cell and triggering its degranulation [109]. This chimeric receptor combines an extracellular portion consisting of an immunoglobulin variable domain fragment (scFv) specific for a surface epitope expressed on the tumor cells with a cytoplasmic portion made of activation and costimulatory domains of key proteins involved in T-cell activation such as CD3, 4-1BB, OX40, or CD28 [110]. These features enable CAR T cells to overcome some of the immunosuppressive cues present in the tumor microenvironment.

Several CAR T cells are currently under development to treat glioblastoma. Target antigen selection is critical, and selected targets’ expression should be be specific to GBM or shared only with nonessential organs. Common targets notably include the neoantigens IL-13R-α2 [111,112], EGFRvIII [113], bispecific receptors to cytomegalovirus (CMV), and HER2 [114]. A case report has shown dramatic radiographical response, with shrinkage of all lesions by 77–100% in a patient treated with repeated intracranial injection of IL-13R-α2 CAR T cells. However, 7.5 months after the first CAR T cell injection, the tumor relapsed [112]. Interestingly, post-treatment biopsies revealed a loss of expression of IL-13R-α2, which seems to indicate that immunoediting resulted in immune escape. Prior to that, the same group reported a phase I trial demonstrating the safety of the approach with little systemic toxicity [115].

O’Rourke et al. reported a pilot trial (NCT02209376) in which they treated 10 patients with recurrent EGFRvIII-positive GBM with a single IV dose of EGFRvIII-specific CAR T cells [113]. This trial showed no clinical benefit, but interestingly, post-treatment biopsies revealed that glioblastoma lesions were infiltrated by CAR T cells, demonstrating their ability to cross the BBB. Again, evidence of immunoediting was observed in five out of seven patients who had surgical resection of their tumor following CAR T cell therapy, but we know that this occurs spontaneously as well. Next to decreased EGFRvIII expression, increased expression of immunosuppressive markers such as IDO1 and PD L1, and elevated numbers of regulatory T cells (Treg) was observed [113].

CAR T cell research for GBM is intense: ongoing CAR T cell clinical trials in GBM include EGFRvIII (NCT01454596, NCT02209376, NCT02844062, and NCT03283631), ephrin type-A receptor 2 (EphA2) (NCT02575261), HER2 (NCT02442297, NCT01109095, and NCT03389230), IL-13Rα2 (NCT02208362), and PD-L1 (NCT02937844). Furthermore, novel potential targets are emerging in preclinical models such as CD70 [116], IL7 receptor [117], and podoplanin (PDPN) [118].

Clinical trials published to date demonstrate that CAR T cells do infiltrate GBM tumors and can eliminate significant tumor volumes, but they also demonstrate the great adaptivity of GBM, enabling it to escape immune attack. Heterogeneous antigen expression, immunosuppressive tumor microenvironments, and immunoediting are major barriers that will need to be overcome in order for CAR T cells to be effective. These challenges faced by CAR T cells are common when tackling solid tumors, and are not restricted to GBM. CAR T cells are approved for highly clonal hematological malignancies lacking tumor microenvironments, but their proof of efficacy in solid tumors has yet to come [119]. Overcoming these challenges will most certainly be achieved by modifying the CAR T cells’ design and using multiple CAR T cells to target many antigens simultaneously [120]: Recently, a trivalent CAR T cell targeting three glioma antigens (IL13Rα2, HER2, and EphA2) was designed, virtually recognizing 100% of GBMs [121]. These developments in CAR T cell design combined with other immunotherapy strategies described above, such as checkpoint inhibitors or oncolytic viruses, might be the key to success in GBM immunotherapy.

### 5.5. Identified Limits and How to Overcome Them

To date, none of the immunotherapeutic approaches has convincingly demonstrated a significant clinical benefit in a randomized phase III trial. This may be linked to a number of factors, including tumor-limiting factors: As illustrated in the ICP section, GBMs are known to show a low mutational burden, providing few therapeutic targets for the immune system. Moreover, the antigen targets may be selectively downregulated in cancer cells, or may only be expressed on a subset of cancer cells that may not be relevant for the survival of the tumor. Moreover, it is possible that GBM has been subjected to significant immunoediting during its development, resulting in a highly immunoevasive and suppressive tumor [122,123]. As mentioned, GBM’s tumor microenvironment (TME) is particularly immunosuppressive, and its targeting might unleash antitumor immune responses. TAM and Tregs are major constituents of the TMEs, and targeting and reprogramming them has been investigated by CSF-1 (NCT02526017), TGFβ (NCT02423343), and IDO (NCT02052648) inhibition, but without clinical benefit so far [37,38]; however, this approach might be useful combined with checkpoint inhibitors or CAR T cells, as it has been shown in murine models [39]. It is very likely that not all tumors will be equally susceptible to immunotherapy, and it will be key to identify which patients are more susceptible to respond to immune approaches. For instance, in the subgroup of patients that presented with responding tumors or stable disease for over 6 months to ICP, there was a clear overrepresentation of patients harboring protein tyrosine phosphatase nonreceptor type 11 (PTPN11) and B-raf murine sarcoma (BRAF) mutations [16]. It is also possible that selecting for patients with higher tumor mutation loads may provide an increased repertoire of neoantigens that may improve the success of immune therapeutic approaches. In colorectal cancer, patients with hypermutated phenotypes linked to mutations in DNA mismatch repair (MMR) genes demonstrated a disease control rate of 90% under ICP treatment, compared to 11% in non-hypermutated patients [124]. In regard to recurrent GBM, it is important to point out that temozolomide induces far fewer mutations than those caused by defects in MMR genes [21,24].

Further, the normal function of the immune system of patients with GBM may be hampered by the use of required supportive treatments such as steroids, or by the myelotoxic effects of chemotherapies. Selecting patients that are not expected to require steroids, at least during the first phases of treatment, may be critical. In patients with newly diagnosed GBM, it might also be critical to consider neoadjuvant immunotherapy prior to radiation therapy and temozolomide treatment, and potentially even before neurosurgical resection of the lesion. This point is highlighted by the results observed in a phase II randomized trial that included 16 + 19 patients showing that administration of the ICP pembrolizumab prior to resection significantly improved OS compared to patients treated in the adjuvant setting only [125].

The choice of the local treatment strategy might also be important with subsequent immunotherapy in mind: preclinical data have demonstrated that localized treatment modalities, especially stereotactic radiotherapy, do synergize with immunotherapy [40,126,127] by releasing tumor antigens. This rationale is at the basis of two ongoing clinical trials combining stereotactic radiotherapy with PD-1 inhibition in GBM (NCT02648633 and NCT02866747), as well as a series of clinical trial combining different antigen-releasing local treatments (local chemotherapy, oncolytic viral therapy, and laser ablation) with immune stimulation (NCT02311582, NCT01811992, NCT02197169, NCT02798406, and NCT02576665).

An adequate study design is also essential to ensure that maximal information can be gained from every clinical trial, and that patients are not exposed to futile treatments. 

Given the complexity of many immunotherapy approaches (for instance, required time to prepare for individualized treatments, tumor size limits, or requirements for absence of steroid use), there is an important risk of selection bias regarding patients that are included in immunotherapy clinical trials that can often not be accounted for when analyzing historical controls. It therefore appears essential that all phase II trials include a randomization process to provide adequate controls and offer the confidence that the observed efficacy endpoints are truly meaningful. This is essential to expedite the process of selecting the immunotherapy approaches that are most likely to succeed, and to minimize the exposition of patients to inefficacious treatments.

Finally, it is very likely that a single immunotherapy approach will not be sufficient to elicit a strong enough response. For this reason, combination therapies are being actively investigated, with the hope to overcome the immunosuppressive environment in GBM and the tumor escape mechanisms. For example, upregulation of alternative immune checkpoint, such as TIM-3 following anti-PD-1 treatment [128], provides a rational for double-checkpoint inhibition with long-term tumor control in preclinical models [127].

## 6. Conclusions

Immunotherapy has revolutionized cancer care. Spectacular responses have been observed across various tumor histologies, in particular with checkpoint inhibitors and CAR T cells, which have been approved as a first-line treatment for certain cancers. However, across numerous studies, GBM has demonstrated strong resistance mechanisms against all phases of the antitumor immune attack. Intrinsic resistance mechanisms and its location beyond the blood–brain barrier prevent the initiation of an immune response, while adaptive resistance inhibits tumor-infiltrating T cells. Immunotherapy clinical trials in GBM have demonstrated very little benefit, if any, with the few patients responding not showing durable responses.

A number of factors may explain the negative findings obtained so far: During tumor development, GBM may have been subjected to significant immunoediting, resulting in a highly immunosuppressive and evasive phenotype [123]. So far, there is little understanding of the intrinsic tumor properties that render them resistant or susceptible to immunotherapy. Clinical characteristics, such as steroid requirement, size of the lesion, and status of resection may play a key role in determining whether a patient will respond to immunotherapy. The timing of immunotherapy may also be key. One could speculate that neoadjuvant administration of immunotherapy treatments may stimulate a more robust immune response than concomitant administration with temozolomide and radiation therapy. It will also be essential to investigate the molecular determinants in the tumor and its microenvironment that must be targeted or that may predict a response to immunotherapy. Finally, it is essential to reconsider the design of clinical trials evaluating immunotherapeutic approaches to quickly gain information about the potential clinical impact of these approaches. However, recent data tend to show that combining immunotherapeutic approaches might allow researchers to finally fulfill the medical need for GBM patients.

## Figures and Tables

**Figure 1 cancers-13-03721-f001:**
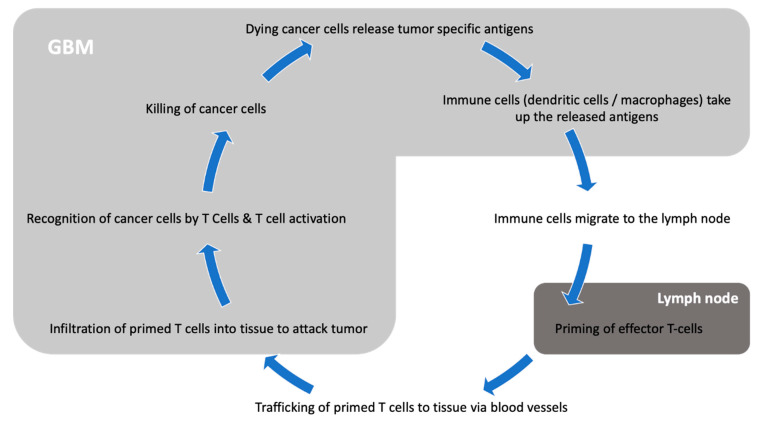
Interaction between GBM and the immune system.

**Figure 2 cancers-13-03721-f002:**
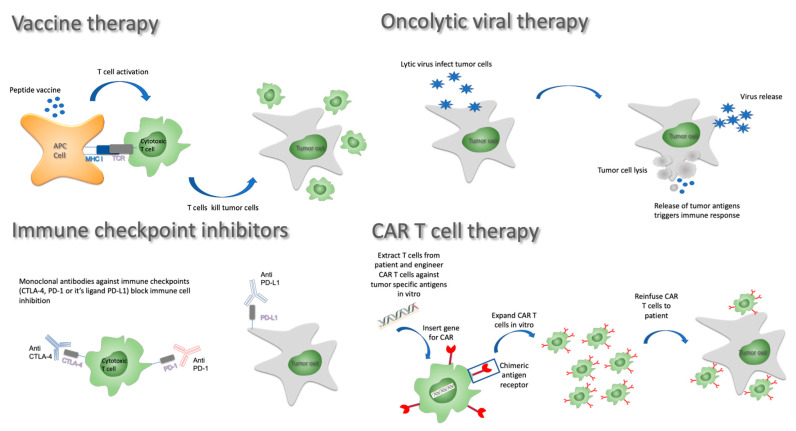
Immunotherapy approaches against GBM.

**Table 1 cancers-13-03721-t001:** Relevant ongoing clinical trials addressing immunotherapies in GBM.

Clinical Trial Name	Description	Phase of Trial	Nb of Participants	Primary Outcome Measure	mOS	mPFS	Significant Results	Demonstrated Immune Response	Comments
**Vaccines**									
*Phase III trials*									
NCT01480479 (ACT IV)	Rindopepimut + TMZ in newly diagnosed EGFRvIII positive patients	3	745	OS	20.1 mo	8.0 mo	No difference in OS (2.1 vs. 20.0 mos) and PFS (8.0 vs. 7.4 mos)	Robust systemic antigen-specific antibody response	Subgroup analysis suggests a modest advantage in patients with residual tumors (≥2 cm^2^)
NCT00045968 (DCVax-L)	DCvax-L in newly diagnosed GBM following resection	3	348	PFS	23.1 mo	not yet available	So far, only OS result of the combined arms reported	Not reported (yet)	No clear indication of positive effect/1599 patients screened and 348 patients included
*Randomized trials*									
NCT01280552 (ICT-107)	Double-blind, randomized study of ICT-107 with maintenance TMZ in newly diagnosed GBM	2	124	OS	17.0 vs. 15.0 mos (HR: 0.87; *p* = 0.58)	11.2 vs. 9.0 mos (HR = 0.57, *p* = 0.58)	No difference in OS, PFS in favor of treatment arm	Robust systemic response	Pts in the HLA-A2 subgroup showed increased ICT-107 activity immunologically with a tendency for improved clinical outcome
NCT03018288 (HSPPC-96)	Double-blind, randomized study of RT + TMZ and pembrolizumab +/− HSPPC-96 vaccine in newly diagnosed GBM	2	108	1 year OS			Ongoing study, estimated completion 01/2025		
*Non-randomized trials*									
NCT02149225 (GAPVAC)	GAPVAC1 and 2, GM-CSF and Poly-ICLC and TMZ in newly diagnosed GBM	1	16	AEs	29 mo	12 mo	Able to generate a strong and lasting immune response	Unmutated APVAC1 antigens elicited sustained responses of central memory CD8+ T cells. APVAC2 induced predominantly CD4+ T-cell responses of T helper 1 type against predicted neoepitope	
NCT02924038	Varlimumab (CDX-1127) + IMA950/polyICLC in newly diagnosed GBM	1	30	Aes, CD4+, CD8+, and T-cell responses			Ongoing study, estimated completion 12/2022		Varlimumab (CDX-1127) is an anti-CD-27 antibody that activates T cells
NCT02287428 (NeoVax)	Personalized neoantigen cancer vaccine (neoVax) + RT + pembrolizumab in newly diagnosed GBM	2	56	Aes; no. of patients with actionable peptides; no. of pts able to recieve post-RT vaccine therapy			Ongoing study, estimated completion 01/2025		
NCT02287428 (NeoVax)	Personalized neoantigen cancer vaccine (neoVax) + RT in newly diagnosed GBM	1b	8	Safety and feasibility	16.8 mos	7.6 mos	Neoantigen selection is feasible and induces immune response	Neoantigen-specific T cells from the peripheral blood could migrate into an intracranial glioblastoma tumour	
NCT02960230	H3.3K27M peptide vaccine in children with newly diagnosed DIPG/gliomas	1	29	Aes; overall OS at 12 months			Ongoing study, estimated completion 01/2023		
**Oncolytic viral therapies**									
*Phase III trials*									
NCT02414165 (Toca 5)	Toca 511 (retroviral replicating vector encoding cytosine deaminase + Toca FC (flucytosine) vs. lomustine, TMZ, or bevacizumab in recurrent HGG	2/3	403	OS	11.1 mos in treatment arm vs. 12.2 in control arm (HR = 1.06, *p* = 0.6154)		Stopped prematurely for lack of efficacy		Data available only from company communication
*Nonrandomized trials*									
NCT01470794	Toca 511 (retroviral replicating vector encoding cytosine deaminase + Toca FC (flucytosine) in recurrent HGG	1	58	MTD, dose-limiting toxicities	14.4 mos		Durable complete responses were observed		
NCT1491893 (PVSRIPO)	Recombinant nonpathogenic polio-rhinovirus chimera (PVSRIPO) in reccurent HGG	1	61	MTD, dose-limiting toxicities	12.5 months (95% CI, 9.9 to 15.2)		21% long-term survivors at 36 months		
NCT02197169 (TARGET-I)	DNX-2401 ± interferon gamma (IFN-γ) for recurrent glioblastoma	2	27				No benefit with the addition of IFN/IFN poorly tolerated		Data available from ASCO poster only
NCT00805376 (DNX-2401)	DNX-2401 (conditionally replication-competent adenovirus) +/− surgery in recurrent HGG	1	37	MTD	9.5 mos		Long-term survivors reported	Treatment induced tumor infiltration by CD8+ and T-bet+ cells	
NCT02798406 (CAPTIVE)	DNX-2401 (conditionally replication-competent adenovirus) + pembrolizumab in recurrent GBM	2	49	Objective response rate			Ongoing study, expected completion 08/2023		
NCT02986178	Recombinant nonpathogenic polio-rhinovirus chimera (PVSRIPO) in recurrent malignant glioma	2	122	Objective radiological response rate at 24 and 36 months			Ongoing study, expected completion 07/2021		
NCT03896568 (Ad5-DNX-2401)	Ad5-DNX-2401 (oncolytic adenovirus) in bone marrow human mesenchymal stem cells in recurrent HGG	1	36	MTD			Ongoing study, estimated completion 05/2022		
NCT01956734 (DNX2401)	DNX-2401 + temozolomide in recurrenct glioblastoma	1	31	Nb of participants with AEs			Study completed 2018, no info available		
NCT02026271	Ad-RTS-hIL-12 with veledimex in recurrent HGG	1	38	Safety and tolerability of varying doses of intratumoral Ad-RTS-hIL-12 and oral veledimex	12.7 mos		Response correlated with CD8+ (cytotoxic) and FoxP3+ (regulatory) T-cell counts in the peripheral blood		
NCT03330197	Ad-RTS-hIL-12 + veledimex in pediatric subjects With brain tumors including DIPG	1	25	Safety and tolerability			Study ongoing, expected completion 12/2022		
NCT00390299	Carcinoembryonic antigen-expressing measles virus (MV-CEA) in trecurrent glioblastoma multiforme	1	23	DLT					
NCT03294486	Safety and efficacy of the oncolytic virus armed for local chemotherapy, TG6002/5-FC, in recurrent GBM	1	78	DLT					
NCT02457845 (G207)	HSV G207 (oncolytic HSV-1) + RT; children with recurrent HGG	1	18	MTD			Enrollment completed 1/2021, results not yet availabe		
NCT03152318 (rQNestin)	rQNestin34.5v0.2 (oncolytic HSV-1) + cyclophosphamide in recuurent HGG	1	108	MTD			Ongoing study, estimated completion 07/2022		Ongoing study
NCT00390299	MV-CEA (carcinoembryonic antigen expressing measles virus) in recurrent GBM	1	23	MTD, severity of Aes, overall toxicity			Accrual completed		
NCT01301430 (ParvOryx01)	H-1 PV in recurrent HGG	1	18	Safety and tolerability					
NCT03714334	DNX-2440 conditionally replication-competent adenovirus with OX40 ligand (T-cell stimulator) in recurrent GBM	1	24	Treatment-related Aes					
NCT02062827 (M032-HSV-1)	M032-HSV-1 (second-generation oncolytic HSV with IL-12 (immune stimulatory) in recurrent GBM	1	36	MTD					
**Checkpoint inhibitors**									
*Phase III trials*									
NCT02017717 (Checkmate 143)	Nivolumab vs. bevacizumab in recurrent GBM	3	626	OS			OS: 9.5 mo vs. 9.8 mo (NS)		
NCT02617589 (Checkmate 498)	Nivolumab + RT vs. RT + TMZ in MGMT unmethylated newly diagnosed GBM	3	550	OS					
NCT02667587 (Checkmate 548)	Nivolumab + RT-TMZ vs. RT + TMZ in MGMT methylated newly diagnosed GBM	3	693	OS					
*Nonrandomized trials*									
NCT02336165	Durvalumab (MEDI4736) in newly diagnosed and recurrent glioblastoma (5 non comparative arms)	2	159	OS at 12 mos			Ongoing study		
NCT02054806	Pembrolizumab (MK-3475) in advanced solid tumors	1b	26 GBMs	Best overall response	14.4 mos	2.8 mos			
NCT02054806	Pembrolizumab in recurrent GBM	2	26				4% partial responses/48% SD		
NCT02337491	Pembrolizumab alone; pembrolizumab + bevacizumab in recurrent GBM	2	80	Pembrolizumab maximum tolerated dose; pembrolizumab dose-limiting toxicity at 6 mos/PFS			Accrual completed		
NCT2313272	Hypofractionated stereotactic RT + pembrolizumab + bevacizumab in recurrent HGG	1	32	MTD					
**CAR T-Cell therapies**									
*Nonrandomized trials*									
NCT2208362	IL13Ralpha2specific CAR T cells in recurrent HGG	1	92	Aes (grade ≥3)			Ongoing study		
NCT02209376	EGFRvIII CAR T cells in EGFRvIII positive GBMs	1	11	Aes at 2 years			Study prematurely terminated		
NCT01109095	HER2 virus-specific CAR T cells	1	16	DLTs			Ongoing study		
NCT02442297	HER2 CAR T cells	1	28	DLTs			Ongoing study		
NCT00331526	Cellular adoptive immunotherapy in recurrent GBM	2	33	Aes/PFS & OS	12.05 mos				
**Combined approaches**									
*Randomized trials*									
NCT02866747 (STERIMGLI)	Hypofractionated stereotactic radiation therapy ± durvalumab in recurrent GBM (STERIMGLI)	1/2	112	DLT (phase 1)/OS (phase 2)			ongoing study, completion expected 12/2024		
*Nonrandomized trials*									
NCT02960230	H3.3K27M peptide vaccine + nivolumab in children with newly diagnosed DIPG/gliomas	1/2	49	Safety of the vaccine in combination with nivolumab			Ongoing study, estimated completion 01/2023		
NCT02648633	Stereotactic radiosurgery with nivolumab and valproate in patients with recurrent glioblastoma	1	4	Feasability					
NCT02311582	Pembrolizumab + MRI-guided laser ablation in recurrent malignant gliomas	1/2	58	MTD (phase 1)/PFS (phase 2)			Ongoing study, expected completion 12/2024		MLA aims at disrupting the blood–brain barrier
NCT01811992	Combined cytotoxic and immune-stimulatory therapy for glioma	1	19	MTD			Ongoing study, expected completion 04/2021		
NCT01205334 (COGLI)	CMV-specific cytotoxic T cells in recurrent GBM								
NCT02798406 (CAPTIVE)	Combination adenovirus + pembrolizumab to trigger immune virus effects in recurrent GBM (CAPTIVE)	2	49	Objective response rate			Ongoing study, enrollment completed 03/21		
**Modification of the tumor microenvironment**									
*Nonrandomized trials*									
NCT02052648	IDO inhibitor + temozolomide in recurrent HGG	1/2	160	Dose determination and 6-month PFS			Accrual completed		Indoximod is an immunometabolic adjuvant that induces T-cell activity in cancer
NCT02526017	Cabiralizumab in combination with nivolumab in patients with selected advanced cancers (FPA008-003)	1	295	Safety			Accrual completed		Cabiralizumab is a humanized monoclonal antibody directed against the tyrosine kinase receptor colony stimulating factor 1 receptor (CSF1R; CSF-1R)

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
