# Peer review of "Immunotherapy in Glioblastoma: A Clinical Perspective"

_cancers, 2021, doi:10.3390/cancers13153721_

Round 1
Reviewer 1 Report
Title: “ Immunotherapy in glioblastoma: a clinical perspective”
The authors performed a highly interesting review of the existing evidence addressing immunotherapy of glioblastoma (GBM). They comprehensively describe the rationale for immunotherapeutic strategies to the most frequent primary brain tumor in the adulthood and the expectations connected to approach. Thankfully, they also address the limitations of immunotherapy in the field of neurooncology in contrast to general oncology, highlighting the organ specific challenges to this approach. This is an important and professionally written paper, addressing a highly relevant aspect of glioma biology. However, there are a few aspects, which need to be addressed before being published in Cancers.
- In the simple summary abstract, the authors state, that GBM is the „most aggressive brain tumor affecting patients.” This sentence seems somewhat strange, since it is obvious that these tumors actually affect patients. I would suggest re – wording to “affecting adult patients”, since only in the adult population GBM is the most frequent primary brain tumor (in contrast to the pediatric population).
- A prognosis is poor, dismal or limited but not dark. Please rephrase.
- In the introduction the epidemiology of GBM is reported with 48.5%. Is that the population – based incidence rate ? Or the fraction of GBM within all primary brain tumors ? Pls. clarify.
- Also in the introduction, the authors mention the efficacy of immunotherapy in other cancer types. Pls. provide some current citations for this statement.
- provide references to support the highly relevant aspect that rGBM may show a significantly higher mutational burden.
- The aspect of immunosurveillance within the CNS was a matter of long time debate. Pls. provide an illustrative reference that this matter is now settled.
- It appears that reference #42 reporting the results of the ACT IV trial is incorrect. The first author of this paper is not Tran, D.D. (this would be author # 3 on this paper), but Weller, M. Pls. correct that error.
- In chapter 5.3. the authors state “Different dosing regimens were tested but in fine, nivolumab alone showed a greater median OS”, this sentence is difficult to understand. Pls. re-phrase.
- Clearly, this review is comprehensive and describes the current evidence in an exhaustive fashion. The tables in the supplement, summarizing the broad range of clinical trial results literature, are somewhat hard to comprehend and need to be reformatted for more straightforward understanding if possible.
Author Response
Responses to the reviewers
Reviewer 1
Title: “ Immunotherapy in glioblastoma: a clinical perspective”
The authors performed a highly interesting review of the existing evidence addressing immunotherapy of glioblastoma (GBM). They comprehensively describe the rationale for immunotherapeutic strategies to the most frequent primary brain tumor in the adulthood and the expectations connected to approach. Thankfully, they also address the limitations of immunotherapy in the field of neurooncology in contrast to general oncology, highlighting the organ specific challenges to this approach. This is an important and professionally written paper, addressing a highly relevant aspect of glioma biology. However, there are a few aspects, which need to be addressed before being published in Cancers.
We thank the reviewer for his positive comments and will address the issues as highlighted below
- In the simple summary abstract, the authors state, that GBM is the „most aggressive brain tumor affecting patients.” This sentence seems somewhat strange, since it is obvious that these tumors actually affect patients. I would suggest re – wording to “affecting adult patients”, since only in the adult population GBM is the most frequent primary brain tumor (in contrast to the pediatric population).
This has been corrected. The portion affecting patients has been erased
- A prognosis is poor, dismal or limited but not dark. Please rephrase.
Was rephrased and changed to dismal
- In the introduction the epidemiology of GBM is reported with 48.5%. Is that the population – based incidence rate ? Or the fraction of GBM within all primary brain tumors ? Pls. clarify.
We thank the reviewer for this comment. After consideration, as this manuscript does not focus on epidemiology, we will highlight that GBM is the most common primary malignant brain tumor without entering into specifics and have erased this statement
- Also in the introduction, the authors mention the efficacy of immunotherapy in other cancer types. Pls. provide some current citations for this statement.
A reference to the work by Waldman et al was added
- provide references to support the highly relevant aspect that rGBM may show a significantly higher mutational burden.
This is a very important point and we have expanded section 4 to stress this point. The following paragraph was added: “Alkylating agents such as temozolomide, given their intrinsic mechanism of action have been reported to induce recurrences with increased tumor mutational load in about 10% of patients {Wang et al., 2016, Nat Genet, 48, 768-76}. These hypermutant recurrent gliomas may increase this tumor adaptability, but may also present unique molecular vulnerabilities with the development of multiple additional tumor specific neoantigens.”
- The aspect of immunosurveillance within the CNS was a matter of long time debate. Pls. provide an illustrative reference that this matter is now settled.
We referred to the review article by Rustenhoven and Kipnis, Nature, 2019 that provides an excellent and complete overview of this particular issue.
- It appears that reference #42 reporting the results of the ACT IV trial is incorrect. The first author of this paper is not Tran, D.D. (this would be author # 3 on this paper), but Weller, M. Pls. correct that error.
corrected
- In chapter 5.3. the authors state “Different dosing regimens were tested but in fine, nivolumab alone showed a greater median OS”, this sentence is difficult to understand. Pls. re-phrase.
This was rephrased to: Various combinations of CPIs and dosing regimens were tested but, in fine, nivolumab alone (3mg/kg body weight every 2 weeks) showed a greater median OS (10.4 months vs 9.2 and 7.3 months) than the combination regimens and less toxicity.
- Clearly, this review is comprehensive and describes the current evidence in an exhaustive fashion. The tables in the supplement, summarizing the broad range of clinical trial results literature, are somewhat hard to comprehend and need to be reformatted for more straightforward understanding if possible.
We thank the reviewer for his positive feedback. Regarding the table, we believe that it presents an important source of comprehensive information for readers that are particularly interested in this topic. We would like to follow the recommendation of reviewer 2 who suggested to add it to the main manuscript.

Reviewer 2 Report
To the authors:
The authors, Desbaillets et al., summarized and discussed the latest finding of glioblastoma treatment approaches. In general, the review is well written and clear. There are some concerns, which should be addressed:
Major concerns:
- The table should be in the manuscript, not “Supplementary data”.
- Additional figure, illustrating and summary of “immunotherapy strategies”.
- Immune evasion is underestimated.
Mainor
- Page 1, 1. Introduction: First word should be glioblastoma multiforme (GBM).
- Page 2, first paragraph: “…where they stand…”, expression of they should be avoided.
- Page 3, 4. Paragraph: “…or flank did not”, expression is confusing
- Page 3: figure legend for figure 1 is missing.
- Page 5, first paragraph: “epitope spreading” should be explained. “immune memory” which cells are meant, memory T cell or memory B cells?
- Page 5, 5.1. Vaccine approaches, 2. Paragraph, end: Reference is missing
- Page 6, 2. Paragraph: “… express this abnormal antigen” reference is missing.
- Page 8, 3. Paragraph: “didn’t”, this expression need to be avoided.
- Page 12, 2. Paragraph: “is be blamed here”, please rewrite.
- Generally, too many “unfortunately” are used, this word should be even avoided completely.
Author Response
Reviewer 2
The authors, Desbaillets et al., summarized and discussed the latest finding of glioblastoma treatment approaches. In general, the review is well written and clear. There are some concerns, which should be addressed:
Major concerns:
- The table should be in the manuscript, not “Supplementary data”.
The table is now included in the main manuscript
- Additional figure, illustrating and summary of “immunotherapy strategies”.
This figure was added
- Immune evasion is underestimated.
We fully agree with this comment from the reviewer. To highlight this part, we have renamed section 4 of our manuscript and expanded the section.
Minor
- Page 1, 1. Introduction: First word should be glioblastoma multiforme (GBM).
Corrected
- Page 2, first paragraph: “…where they stand…”, expression of they should be avoided.
Replaced by “current status”
- Page 3, 4. Paragraph: “…or flank did not”, expression is confusing
The sentence was modified to: “This phenomenon was specific to the brain, as it was not observed when identical tumors were implanted in the lungs or flank.”
- Page 3: figure legend for figure 1 is missing.
The legend: Figure 1: interaction between GBM and the immune system was added
- Page 5, first paragraph: “epitope spreading” should be explained. “immune memory” which cells are meant, memory T cell or memory B cells?
Epitope has been explained by adding: (a process whereby epitopes that are distinct from and non-cross-reactive with an inducing epitope become major targets of the ongoing immune response). B&T cell Immune memory was added
- Page 5, 5.1. Vaccine approaches, 2. Paragraph, end: Reference is missing
The reference to the review by Weller et al, Nat Rev Neurol, 2017 has been added
- Page 6, 2. Paragraph: “… express this abnormal antigen” reference is missing.
The references were added: Brennan et al., 2013, Cell, 155, 462-77}{Weller et al., 2014, Int J Cancer, 134, 2437-47}
- Page 8, 3. Paragraph: “didn’t”, this expression need to be avoided.
The sentence was changed to: Although the addition of IFN-γ addition did not improve survival,
- Page 12, 2. Paragraph: “is be blamed here”, please rewrite.
The sentence was rewritten in the following manner: “which seem to indicate that immunoediting resulted in immune escape”
- Generally, too many “unfortunately” are used, this word should be even avoided completely.
This has been changed throughout the manuscript
Round 2
Reviewer 2 Report
R1: To the authors:
The authors, Desbaillets et al., provided a revised version of the manuscript. There are some minor errors which need to be addressed:
- The new figure, Figure 2, MHC I molecules are recognized by cytotoxic T lymphocytes (CTLs). Not MHC II as indicated in the figure.
- Immune evasion mechanisms are still underestimated.
- Page 5, definition for epitope spreading is not clear rather misleading.
Author Response
The authors, Desbaillets et al., provided a revised version of the manuscript. There are some minor errors which need to be addressed:
- The new figure, Figure 2, MHC I molecules are recognized by cytotoxic T lymphocytes (CTLs). Not MHC II as indicated in the figure.
We thank the reviewer for pointing this mistake out. It has been corrected
- Immune evasion mechanisms are still underestimated.
We extended the discussion on mechanisms of immune evasion in section 4: “Mechanisms of immune evasion have been identified such as prevention of entry of immune cells by an intact blood brain barrier, immune suppression by the tumor mi-croenvironment, reshaping of the immune microenvironment by regulatory T cells and tumor-associated macrophages and interfering with key immune pathways for example with immune checkpoint receptor expression. These mechanisms of immune evasion used by GBM are the subject of intensive research and an in-depth review goes beyond the scope of this article but have been thoroughly reviewed in {Razavi et al., 2016, Front Surg, 3, 11}.”
- Page 5, definition for epitope spreading is not clear rather misleading.
We tried to clarify the definition and changed it to: a process whereby the anti-tumor immune response broadens and novel epitopes distinct from and non-cross-reactive with the initially targeted tumor epitope become additional targets